# Characterization of *Paulownia elongata x fortunei* (BIO 125 clone) Roundwood from Plantations in Northern Italy

Irene Criscuoli [1,2], Michele Brunetti [3,*] and Giacomo Goli [1]

1   DAGRI-Department of Agriculture, Food, Environment and Forestry, University of Florence, 50145 Florence, Italy
2   CREA-Research Centre for Agriculture and Environment, 50125 Florence, Italy
3   CNR IBE-Institute of Bioeconomy, 50019 Sesto Fiorentino, Italy
*   Correspondence: michele.brunetti@cnr.it

**Abstract:** The growth performance and technological quality of roundwood from a *Paulownia elongata x fortunei* hybrid (BIO 125 clone) was assessed in three plantations in Northern Italy. Dendrometric features (diameter at several heights, volume, and growth rate) and defects for industrial use were assessed on 20 standing trees and four logs per plantation. Compared to previously published literature, Paulownia trees have shown a high growth rate during the first three years after coppicing. Growth rate sharply decreased starting from the fourth year, suggesting an increased competition between trees and the need for greater planting spacing. At the end of the first year of growth, trees were pruned up to a height of 5 m, allowing the production of defect-free and high-quality roundwood. Log features were assessed according to European standards EN-1309-2:2006 and EN 1309-3:2018 and then compared with the EN 1316-2:2012 standard for poplar roundwood quality. Paulownia wood has shown to be of excellent quality, ranking in the best class (Po-A) for all parameters except diameter. A larger diameter could be easily obtained with longer growth cycles or greater planting spacing. A relevant problem for the industrial exploitation of Paulownia small-diameter logs would be the large empty pith that could drastically reduce timber yields.

**Keywords:** Paulownia; growth; timber; defects; BIO 125 clone

## 1. Introduction

Paulownia is a hardwood, deciduous, fast-growing tree, indigenous to China [1]. Due to its ability to adapt to a broad range of climatic and edaphic conditions, today Paulownia is commonly found in several Asian countries, Europe, Australia, and northern and central America [1,2]. It has also been shown to adapt to arid environments [3] and mining lands to be reclaimed [4,5].

Paulownia wood is characterized by lightness (low density), strength, working ease, stability under rapid drying, and lack of major defects. These characteristics have made Paulownia wood desirable and marketable since ancient times [2,6,7]. For instance, in China it has been used for 2600 years [1] where it is used, as in Japan, for the manufacture of traditional musical instruments, jewelry, and furniture. Paulownia is nowadays spread worldwide, not only for industrial applications (furniture, paper, and energy) but also for ornamental purposes [6], apiculture, and in the medical industry for the treatment of diseases such as bronchitis, asthma, and diabetes [8]. Paulownia cultivation is usually based on short-rotation silviculture techniques including coppicing, pruning, fertilization, and irrigation [6,9–11].

Paulownia is a genus belonging to the *Paulowniaceae* family (previously classified in the *Scrophulariaceae*) [1,2] consisting of nine species including *P. elongata*, *P. fortunei*, *P. kawakamii*, and *P. tomentosa* [1]. Paulownia growth, survivorship, and habitat requirements differ depending on the species [1], origins, and clones both at the nursery stage [12] and early plantation stage [13] as well as in elder managed [1,14] and unmanaged

stands [1,15]. For instance, [15] showed that in the Appalachian Mountains (NC, USA), 9 years after planting, *Paulownia fortunei* had a significantly smaller diameter at breast height (DBH), diameter at ground level, and total height compared to *P. tomentosa* and *P. elongata*, while [1] observed, on average, an 18–36% larger trunk timber volume in *P. fortunei* compared to *P. elongata* trees grown in several Chinese provinces. Paulownia species, hybrids, and clones also differ in physical and mechanical properties, such as density, modulus of elasticity, shrinkage, compression, and bending strength [16,17]. This variety highlights the need for evaluating growth performance and wood quality for all Paulownia species, hybrids, and clones in different pedoclimatic conditions, this information being crucial for plantation managers.

In the present work, we focus on a *Paulownia elongata x fortunei* hybrid, clone BIO125, a sterile and non-invasive interspecific hybrid which is cultivated by the majority of the 65 agricultural companies that grow Paulownia in Emilia-Romagna, Northern Italy, on a surface of 130 ha. The clone was developed in Spain where is grown other than Italy.

There is not a vast amount of literature on *P. elongata x fortunei* hybrids, and to our knowledge no scientific paper has been published about clone BIO125. As with other Paulownia species and hybrids, *P. elongata x fortunei* is cultivated for production of timber, biomass, and pulp [18], and lately its leaves and petioles have been proposed for the production of components with antioxidant and antiradical properties [19–21]. The clone in vitro 112$^{®}$ (*Paulownia elongata x fortunei*) was selected in Spain to resist temperature extremes (from −25 to +45 °C) and to grow at a faster rate compared to other Paulownia species and hybrids (25–30 cm DBH and a height of 15–20 m within 3 years) [12,22], even if some studies have reported worse performances. For example [14] observed that Paulownia x 'Henan 1', a hybrid deriving from an initial cross between *P. elongata* and *P. fortunei*, had a lower survival rate and growth potential compared to *P. fortunei*, and [23] observed no significant differences in the growth performance of *P. tomentosa* and *P. elongata x fortunei* in a greenhouse experiment in Bulgaria. These contrasting results highlight the need to clarify the potential of the *P. elongata x fortunei* hybrid and more specifically the clone BIO125, whose performance in the field has not yet been reported in any scientific publication.

Moreover, [24] highlighted the need to verify the adaptability of *P. elongata x fortunei* to different climatic conditions. The first experimental trials of Paulownia in Italy were performed in the 1970s with *Paulownia fortunei* in central Italy (Lazio) and with *Paulownia tomentosa* in Sardinia, while in the 1990s wider plantations were established in Veneto with more promising results [25]. More recently, [26] showed that *P. tomentosa* had a comparable productivity to *Robinia pseudoacacia* L. and a slightly lower productivity compared to poplar and willow clones grown in the Po river valley (Piedmont), with the advantage that *P. tomentosa* required less tending. For its adaptability to different environmental conditions, in 2016 *P. tomentosa* was also suggested as a replacement for the widely spread poplar clones in northern Italy used in timber and honey production [27]. On the other hand, no specific study nor recommendations have been made in Italy for the *P. elongata x fortunei* hybrid or for the BIO125 clone.

For all these reasons, the scope of the present work is to answer the following research questions: (a) what is the growth potential of *Paulownia elongata x fortunei* (clone BIO125) in Emilia-Romagna, northern Italy? (b) what are the characteristics of the *Paulownia elongata x fortunei* (clone BIO125) roundwood grown in Emilia-Romagna? More specifically, dendrometric parameters and log defects were evaluated on standing and fallen trees in three plantations, with age ranging from 3 to 6 years. Growth performance and log quality were compared with quality standards for poplar (EN 1316-2:2012), another fast-growing broadleaved tree with similar potential use. The rationale behind this study was to verify the performance of the BIO125 clone in terms of growth potential and quality.

## 2. Materials and Methods

### 2.1. Plantation Characteristics

Three Paulownia plantations were selected in the Emilia-Romagna region in northern Italy, within sites belonging to the Paulownia growers' network "PAULOWNIA-Crescere in Rete", a partner of the EAFRD project, PSR Emilia Romagna 2014–2020, n° 5111574. The sites to be analyzed were chosen in order to represent the broadest possible age range (3–6 years old) considering the very recent Paulownia spread in Italy and were grown, as described further in the text, in similar pedoclimatic conditions.

A description of the main experimental areas' characteristics is reported in Table 1 and the geographical location is shown in Figure 1.

**Table 1.** Characteristics of the experimental areas.

| Site Characteristics | Plantation | | |
|---|---|---|---|
| | A | B | C |
| Location | Roncadello di Forlì (FC) | Fratta Terme di Bertinoro (FC) | Libolla di Ostellato (FE) |
| Longitude | 44°16′30.2″ N | 44°09′27.3″ N | 4°44′40.6″ N |
| Latitude | 12°02′42.4″ E | 12°06′15.8″ E | 11°54′13.3″ E |
| Environment | alluvial plain | piedmont plain | deltaic plain |
| Slope (%) | 0.1–0.2 | 0.5–0.8 | 0.05–0.1 |
| Elevation (masl) | 20 | 85 | 2 |
| Soil depth | very deep | very deep | very deep |
| CaCO3 (% in the first 50 cm) | 16 | 18 | 10 |
| pH | 8 | 8 | 8 |
| Texture * | silty loam | silty clay loam | silty clay loam |
| Average annual precipitation (2010–2018) (mm) ** | 724 | 787 | 686 |
| Average annual max. temp. (2010–2018) (°C) ** | 19.9 | 19.2 | 19.5 |
| Average annual min. temp. (2010–2018) (°C) ** | 8.6 | 9.1 | 9.2 |
| Surface (ha) | 1.2 | 9 | 5 |
| Planting | Winter 2016–2017 | May 2016 | 2014 |
| Plant spacing | 3.9 m × 3.9 m | 3.8 m × 3.8 m | 4 m × 4 m |

\* Geological, Seismic, and Soil Service of the Emilia-Romagna Region: (A) https://geo.regione.emilia-romagna.it/cartpedo/scheda_suolo.jsp?id=SMB1 (accessed on 9 August 2022); (B) https://geo.regione.emilia-romagna.it/cartpedo/scheda_suolo.jsp?id=BEL1 (accessed on 9 August 2022); (C) https://geo.regione.emilia-romagna.it/cartpedo/scheda_suolo.jsp?id=BAU1 (accessed on 9 August 2022); \*\* Hydrological, Meteorological and Climate Service of ARPAE Emilia-Romagna, https://dati.arpae.it/dataset/erg5-eraclito (accessed on 9 August 2022).

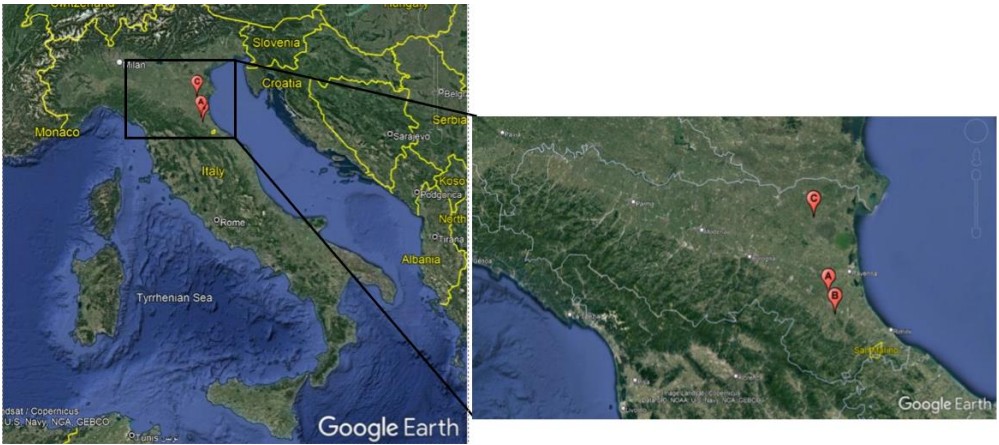

**Figure 1.** Geographical location of the three plantation sites (A, B, and C) analyzed in the present work.

The trees on the three plantations belong to clone BIO 125, a sterile and non-invasive interspecific hybrid of *Paulownia elongata x fortunei*. One year after planting, at the end of winter, stems were cut back to ground level (coppiced). Then, in spring, one out of several shoots was selected to grow while the others were removed from the stump. On the selected shoot, during summer and early autumn, axillary buds were constantly manually removed. At plantation B, farmers were not able to remove the axillary buds above 5 m during summer and early autumn, therefore between 5 and 6 m of height the pruning was completed in winter, when secondary shoots had already become thick branches that were cut with a telescopic pole saw. Other than coppicing and stem pruning, the intensive management regimes also included fertilization and irrigation. Different irrigation and fertilization policies were used in the three areas, but in strongly intensive regimes.

## 2.2. Data Collection

In October 2020 the following parameters were measured at the plantations:

- Dimension and growth: crown insertion height (m); plant total height (m); breast height, large, middle, and top diameter (cm); bark thickness (mm); under-bark log volume (m$^3$); annual ring number and thickness (cm); annual growth rate (cm year$^{-1}$).
- Defects: pith width (mm); eccentric pith (%); buttress height (cm); taper (cm m$^{-1}$); simple and multiple sweep (%); ovality (%); knots (n); rot; scars; wind shakes (m); number of plants broken by the wind (n); sun scald of the rhytidome.
- Mass and volume of fresh small logs: the over-bark volume (m$^3$) and the mass (kg) of short logs were measured to determine roundwood density (kg m$^{-3}$).

The abovementioned parameters were measured on 4 representative plants per plantation that were felled. For a broader characterization of the stand, diameter at breast height, large diameter, sweep, and taper were measured. For every plantation the interior plant height was measured on 20 standing trees, and for plantation C 20 plants at the margin of the plantation were measured for comparison with the interior ones. This comparison was necessary because at plantation C the authors observed during the field campaign an evident difference in growth parameters between the interior plants and those at the edge, and plants at the edge of the plantations were considered as a proxy of the potential growth in case of greater plant spacing. In more recent plantations (A and B), the difference in growth parameters was less evident.

The height of the crown insertion and the total plant height were measured using a Vertex 5 (Haglöf Sweden AB). Diameters were measured with a tree caliper and, if not expressly declared, include the bark.

Bark thickness was measured at the top and at the base of the log, at the end points of two perpendicular diameters (4 measurements for each side of the log). The under-bark log volume was calculated based on the middle diameter as illustrated by the Annex B of the EN 1309-2:2006 standard. The over-bark volume was determined with the same principle by measuring over-bark.

The number of annual rings was counted at the top and the base of the stem, while the average annual growth rate was calculated as:

$$\frac{\text{diameter at breast height (cm)}}{\text{plant age (years)}} \tag{1}$$

The diameter at breast height was chosen in order to avoid the measurement of buttressing in the lower part of the stem.

To perform a more accurate evaluation of the annual growth rate, ring thickness was measured in the laboratory with a digital caliper ($\pm$0.01 mm) on 2 perpendicular radii on 4 wood rounds per plantation retrieved at breast height.

Taper (cm m$^{-1}$) was calculated as indicated by EN 1309-3, with the following equation:

$$\frac{\text{diameter at breast height} - \text{top diameter}}{\text{log lenght above 130 cm}} \tag{2}$$

Simple sweep was calculated as indicated by EN 1309-3, with the equation below:

$$\frac{f}{D} * 100 \tag{3}$$

where f is the sagitta (cm) and D is the middle diameter (cm).

Simple sweep was measured on the 4 felled plants at the three plantations. An extra measure on 20 standing plants at plantation C was performed for a broader characterization of the stand.

Multiple sweep is the cumulative value of simple sweeps observed on the same log, and it was calculated according to EN 1309-3:2018 based on values measured with [3].

Ovality was calculated according to EN 1309-3 on a transverse section cut at a height of 130 cm, as follows:

$$\frac{\text{diameter max} - \text{diameter min}}{\text{diameter max}} * 100 \tag{4}$$

Roundwood density was measured for each sampled tree by removing a 1 m long portion from the bottom of the felled plants. The volume was determined by measuring the circumferences at the two extremities and by calculating the area of an equivalent round surface. The equivalent areas at the two extremities were averaged and multiplied by the length of the log (1 m).

### 2.3. Data Analysis

Data shown in the graphs and tables of the present work represent the average value per plantation ± standard error. Considering that the trees in the three plantations had different ages, most of the parameters are reported in the present work to describe three Paulownia stands, and not to compare them. In the case of parameters normalized for the age of the plant (e.g., average annual growth in different plantations) analysis of variance (ANOVA) was applied to scrutinize for statistically significant differences between plantations or year of growth ($p < 0.05$). When ANOVA highlighted a significant difference, post-hoc individual comparisons were performed with Tukey's HSD test. In the case of comparison between roundwood characteristics in interior plants and plants placed at the edge of plantation C, the *t*-test was applied. When the authors compared the thickness of rings between several years of growth in the same plantation, a generalized mixed linear model was used, considering the tree as a random factor, and then a paired comparison between annual rings was performed, correcting the outputs for multiple comparisons with the Kenward–Roger method. Homoscedasticity and normality were verified before the statistical analysis using the Levene and Shapiro–Wilk tests, respectively. When these assumptions were not verified, data were log-transformed. Statistical analyses were performed using the R software (version 4.2.1) [28].

### 2.4. Quality Assessment

To compare the quality of the logs of Paulownia wood with a well-known fast-growing resource such as poplar wood, largely grown in central-northern Italy and with a well-established value chain, the EN 1316-2:2012 standard was used. This European standard, whose title is "Hardwood round timber–Qualitative classification–Part 2: Poplar", defines four quality grades ranging from Po-A to Po-C, indicating the best and the worst grade respectively. Grades are identified according to thresholds in log characteristics such as diameter, length, number and dimension of knots, sweep, ovality, heart shakes, ring shakes, frost cracks, rot, and insect attacks.

## 3. Results

### 3.1. Dimension and Growth

The crown insertion height was very similar in plantations A (4.9 ± 0.1 m) and C (5 ± 0.1 m) and slightly higher in B (5.4 ± 0.1 m) (Table 2), where the trees were pruned to a greater height.

**Table 2.** Paulownia roundwood characteristics in three plantations (A, B, and C) in Emilia-Romagna, northern Italy. Values are reported as mean ± s.e. "n.s." next a value indicates no significant difference among values measured in the three plantations. When no letters are reported in the cells next to the numbers, it means that no statistical comparison was performed.

| Parameter (Unit) | Plantation | | | | | |
|---|---|---|---|---|---|---|
| | **A** | | **B** | | **C** | |
| Evaluated trees (n) | 4 | | 4 | | 4 | |
| Age (years) | 3 | | 4 | | 6 | |
| Crown insertion height (m) | 4.9 | ±0.1 | 5.4 | ±0.1 | 5 | ± 0.1 |
| Total height (m) | - | | - | | 17.8 | ± 0.3 |
| Diameter at breast height (cm) | 24.3 | ±1 | 23.7 | ±0.2 | 25 | ± 0.5 |
| Large diameter (cm) | 29.2 | ±0.5 | 30.3 | ±0.9 | 31.9 | ± 0.4 |
| Middle diameter (cm) | 20.6 | ±0.7 | 21.5 | ±0.1 | 22.8 | ± 0.4 |
| Top diameter (cm) | 16.9 | ±0.8 | 19.7 | ±0.5 | 20.2 | ± 0.2 |
| Bark thickness at the top of the log (mm) | 3.6 | ±0.2 | 3.5 | ±0.3 | 4.6 | ± 0.7 |
| Bark thickness at the base of the log (mm) | 4.1 | ±0.4 | 3.3 | ±0.3 | 5.3 | ± 0.4 |
| Under-bark log volume ($m^3$) | 0.5 | ±0.0 | 0.6 | ±0.0 | 0.6 | ±0.0 |
| Number of growth rings at the base of the stem | 3 | ±0.0 | 4 | ±0.0 | 6 | ±0.0 |
| Number of growth rings at the top of the stem | 2 | ±0.0 | 3 | ±0.0 | 4.5 | ±0.3 |
| Pith thickness (mm) | 24.9 | ±3.3 n.s. | 23.6 | ±3.8 n.s. | 21.5 | ±0.3 n.s. |
| Eccentric pith (cm) | - | | - | | 2.8 | ±1.3 |
| Buttress height (cm) | 50 | ±4.1 | 43.3 | ±5.8 | 30 | ±12.2 |
| Taper (cm m$^{-1}$) | 2.1 | ±0.1 | 1 | ±0.1 | 1.4 | ±0.9 |
| Ovality (%) | 5.8 | ±1.1 | 3.7 | ±1.8 | 2.1 | ±0.8 |
| Density of fresh roundwood, bark included (kg m$^{-3}$) | 659 | ±49 | 589 | ±12 | 531 | ±94 |

Plant height measured on interior plants was 13.0 ± 1.4 cm for plantation A, 15.7 ± 1.3 cm for B, and 17.8 ± 0.4 m for plantation C (Table 2).

The diameter measured at breast height (DBH) was very similar for the three plantations (A: 24.3 ± 1 cm, B: 23.7 ± 0.2 cm, C: 25 ± 0.5 cm) despite the different ages of the trees (A: 3 years old; B: 4 years old; C: 6 years old) (Table 2). Diameters were also measured on 20 plants at plantation C that were located at the edge of the stand and compared with 20 interior plants (Table 2). The average DBH of the plants located at the edge was significantly greater (30.6 ± 0.7 cm) than the 20 interior plants used as reference (23.5 ± 0.5 cm) (Table 2).

The large diameter of the log was slightly larger in stand C (31.9 ± 0.4 cm) than in stands B (30.3 ± 0.9 cm) and A (29.2 ± 0.5 cm) (Table 2), and at plantation C, the large diameter of the trees grown at the edge (35.2 ± 0.9 cm) was significantly larger than the large diameter of the interior plants (29.2 ± 0.7 cm) (Table 2). In addition, the middle diameter was slightly greater for plantation C (22.8 ± 0.4 cm) compared to B (21.5 ± 0.1 cm) and A (20.6 ± 0.7 cm), as well as the top diameter (C: 20.2 ± 0.2 cm; B: 19.7 ± 0.5 cm; A: 16.9 ± 0.8 cm) (Table 2).

Bark thickness at the base and at the top of the log were very similar for all the plantations (A: top 3.6 ± 0.2 mm, base 4.1 ± 0.4 mm; B: top 3.5 ± 0.3 mm, base 3.3 ± 0.3 mm; C: top 4.6 ± 0.7 mm, base 5.3 ± 0.4 mm), with the highest values observed at plantation C (Table 2). The average value of bark thickness for the three plantations was 4.1 ± 0.3 mm.

The under-bark log volume for plantation A was 0.5 ± 0.0 m$^3$, and at both plantations B and C it was 0.6 ± 0.0 m$^3$ (Table 2).

At the base of the stem, we observed 3 rings for plantation A, 4 rings for plantation B, and 6 for plantation C (as expected based on the age of planting), while at the top of the stem we observed, on average, 2 rings for plantation A, 3 rings for plantation B, and 4.5 for plantation C (in two trees we observed 5 rings and in two others we observed 4 rings).

The average annual growth rate was 8.1 ± 0.3 cm at plantation A, 5.9 ± 0.0 cm at plantation B, and 4.5 ± 0.0 cm at plantation C (Figure 2), where the growth rate was signifi-

cantly higher in the plants grown at the edge of the plantation (5.1 ± 0.0 cm) compared to the interior trees (3.9 ± 0.0 cm) (Figure 3).

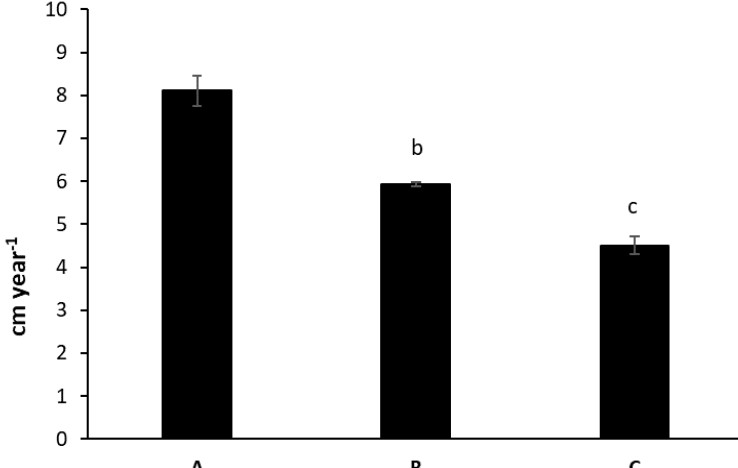

**Figure 2.** Average annual growth rate (cm year$^{-1}$) at the three plantations. Values are reported as mean ± s.e. Different letters at the tops of the bars indicate a significant difference among values measured for the plantations.

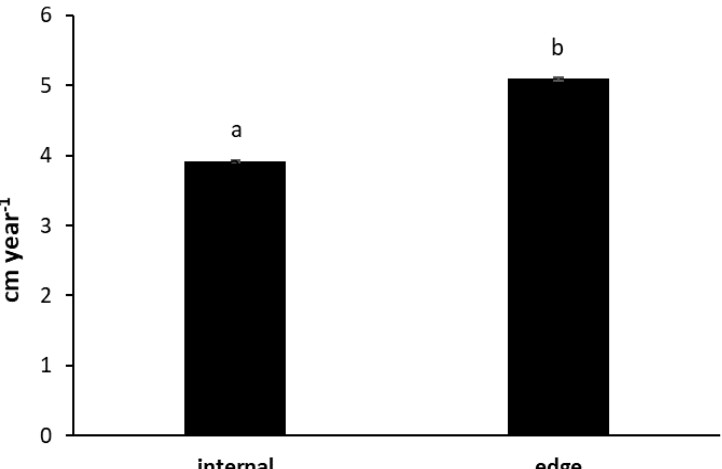

**Figure 3.** Annual growth rate (cm) measured for interior trees and trees at the edge of plantation C. Values are reported as mean ± s.e. Different letters at the tops of the bars indicate a significant difference among values measured for the plantations.

The ring thickness for every year of growth is reported in Figure 4. At plantation A, the first two annual rings (1st year: 41.3 ± 2.8 mm; 2nd year: 36.2 ± 2.2 mm) were significantly thicker than the third and last year (27.6 ± 1.1 mm) (Figure 4). At plantation B, the first-year rings were, on average, 31.3 ± 2.2 mm thick, the second year 39.9 ± 1.9 mm, and the third year 25.1 ± 0.9 mm. The fourth and last annual ring was significantly thinner (12.5 ± 0.5 mm) than the rings of the three years before (Figure 4). At plantation C, we could also observe significantly thicker annual rings in the first two years of growth (1st year: 32.8 ± 1.0 mm; 2nd year: 32.7 ± 2.7 mm) compared to the following years, where the growth rate kept decreasing significantly (3rd year: 18.7 ± 1.4 mm; 4th year: 15.1 ± 1.8 mm, 5th year: 10.6 ± 1.1 mm and 6th year: 6.9 ± 1.1 mm) (Figure 4).

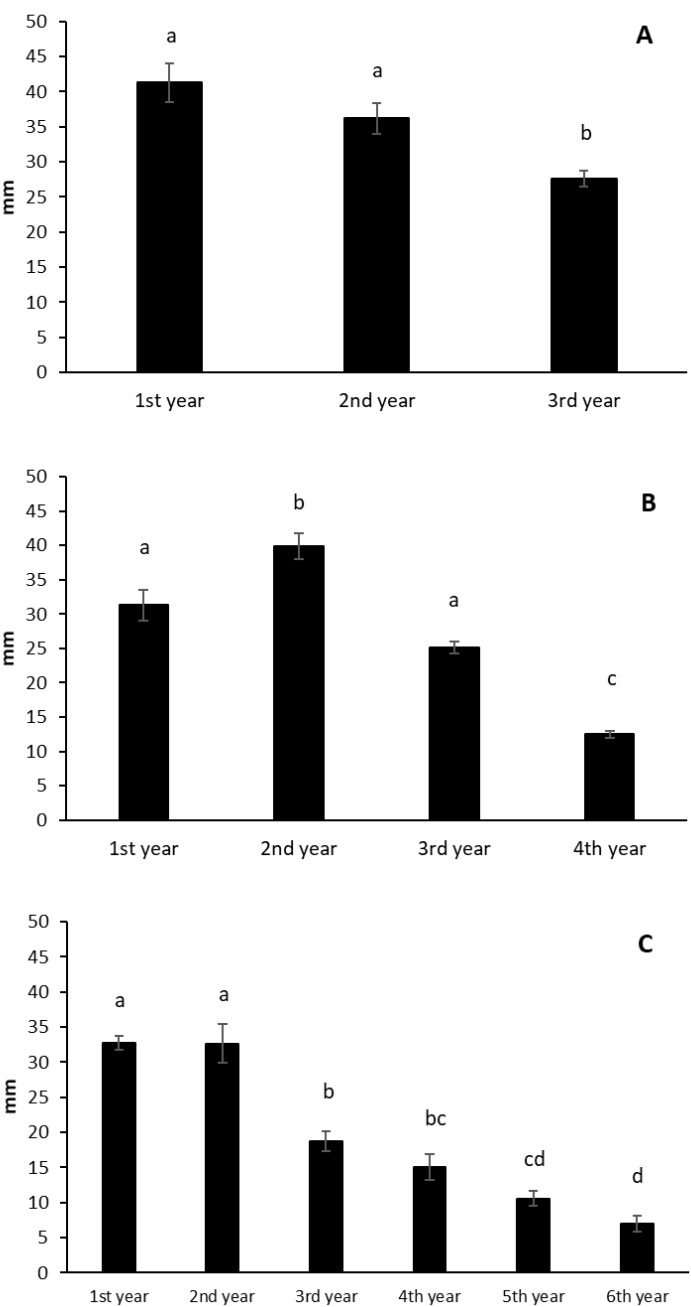

**Figure 4.** Annual ring thickness (mm) at plantations (**A**–**C**). Values are reported as mean ± s.e. Different letters at the tops of the bars indicate a significant difference among values measured for the plantations.

By comparing the ring thickness of every year at the three plantations, we observed a significantly thicker first year ring at plantation A (41.3 ± 2.3 mm) compared to plantations B (31.3 ± 2.2 mm) and C (32.8 ± 1.0 mm). During the second year, the growth rate was statistically comparable for the three plantations (A: 36.2 ± 2.2 mm; B: 39.9 ± 1.9 mm; C: 32.7 ± 2.7 mm), while in the third year, growth was significantly lower for plantation C (18.7 ± 1.4 mm) compared to A (27.6 ± 1.1 mm) and B (25.1 ± 0.9 mm) (Figure 5). From a visual comparison between interior plantings and trees placed at the edge of plantation C (Figure 6), it was evident that after the second year, growth ring thickness was greater in the edge plants than in the interior plants (not measured data). This is also consistent with the greater large, middle, and top diameters of the trees located at the edge of plantation C compared to the interior trees.

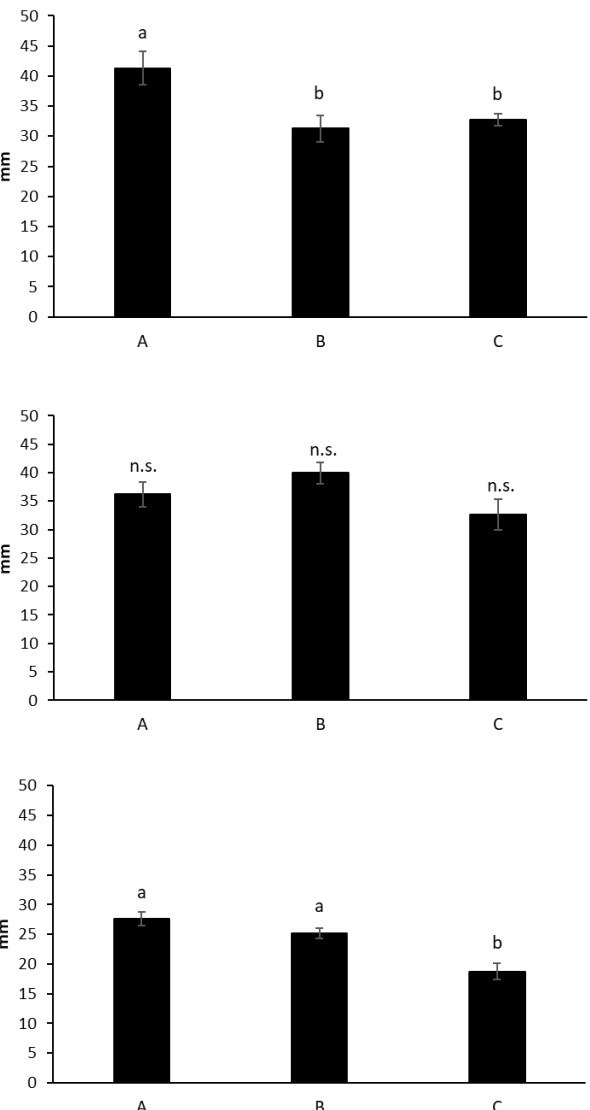

**Figure 5.** Annual ring thickness (mm) in the first three years of growth for plantations A, B, and C. Values are reported as mean ± s.e. Different letters at the tops of the bars indicate a significant difference among values measured for the plantations.

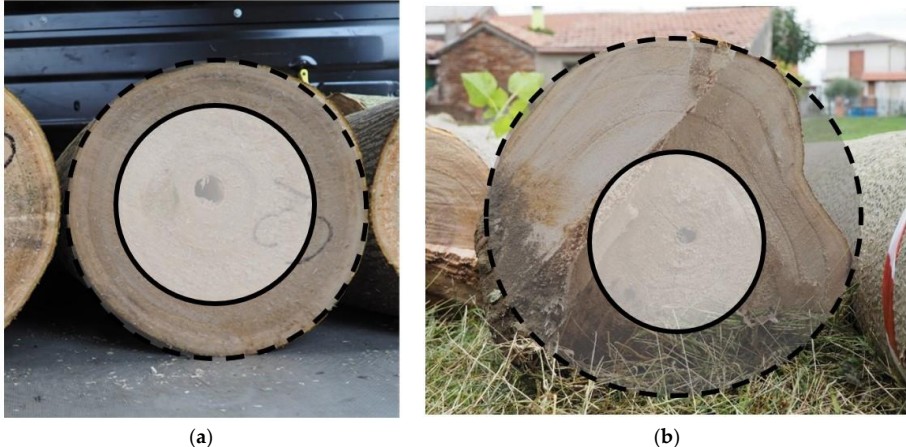

(**a**)　　　　　　　　　　　　　　　　　(**b**)

**Figure 6.** Thickness of the first two annual rings (white area delimited by the black continuous line) and of the subsequent four annual rings (light grey area delimited by the dashed line) observed in one interior tree (**a**) and in one tree grown at the edge (**b**) of plantation C.

### 3.2. Defects

It is well known that Paulownia trees have an empty pith. The pith thickness was not statistically different for the three plantations (A: 24.9 ± 3.3 mm, B: 23.6 ± 3.8 mm, C: 21.5 ± 0.3 mm). Eccentric pith was observed in only two trees out of four at plantation C (2.8 ± 1.3 cm), while it was not observed at the other two plantations (Table 2).

At plantation A, on average, we observed 50 ± 4.1 cm of buttress height from the collar; in stand B, 43.3 ± 5.8 cm, and for C, 30 ± 12.2 cm (Table 2).

Taper values were the greatest at plantation A (2.1 ± 0.1 cm m$^{-1}$), followed by plantations C (1.4 ± 0.3 cm m$^{-1}$) and B (0.9 ± 0.1 cm m$^{-1}$) (Table 2). Taper measured on interior trees located inside plantation C was not significantly different from trees located at the edge of the stand (interior: 1.4 ± 0.1 cm m$^{-1}$, edge: 1.9 ± 0.0 cm m$^{-1}$, Table 3).

**Table 3.** Paulownia roundwood characteristics measured for 40 trees, 20 interior and 20 located at the edge of plantation C. Different letters in the cells indicate a statistically significant difference among values measured at the three plantations, while "n.s." indicates no significant difference.

| Parameter (Unit) | Tree Location at Plantation C | | | | | |
|---|---|---|---|---|---|---|
| | Interior | | | Edge | | |
| Diameter at breast height (cm) | 23.5 | ±0.5 | a | 30.6 | ±0.7 | b |
| Large diameter (cm) | 29.2 | ±0.7 | a | 35.2 | ±0.9 | b |
| Taper (cm m$^{-1}$) | 1.4 | 0.1 | n.s. | 1.9 | ±0.0 | n.s. |

Simple sweep was observed in two felled trees out of four at plantation A (10 ± 2.5) and in one plant out of four at plantation B (7.5) (Table 4). No simple sweep was observed at plantation C on the four felled trees (Table 4), but a broader characterization of the stand highlighted that two plants out of 20 showed a simple sweep with an average value of 16.3 ± 0.3. Multiple sweep was observed only on one tree, at plantation B, with a value of 21.6 (Table 4).

**Table 4.** Paulownia roundwood characteristics measured at plantations A, B, and C.

| Parameter (Unit) | Plantation | | |
|---|---|---|---|
| | A | B | C |
| Evaluated plants (n) | 4 | 4 | 4 |
| Plants with simple sweep (n) | 2 | 1 | 0 |
| Average simple sweep (%) | 10 | 7.5 | 0 |
| Plants with multiple sweep (n) | 0 | 1 | 0 |
| Average multiple sweep (%) | 0 | 21.6 | 0 |
| Plants with eccentric pith (n) | 0 | 0 | 2 |
| Average eccentric pith (cm) | 0 | 0 | 2.75 |

Ovality at plantation A was, on average, 3.7 ± 1.8%, at plantation B, 2.1 ± 0.8%, and at plantation C, 5.8 ± 1.1% (Table 2).

At plantations A and C, trees did not show any knots nor traces of knots on the stem, while at plantation B, nine covered and sound knots per plant were observed on the four felled plants. At plantation B, knots were concentrated in the top meter of the stem because of the late pruning conducted at this plantation.

We did not observe any rot at the top nor at the base of the stem in any of the felled plants at the three plantations, nor unhealed scars where trunks were coppiced one year after planting.

At all the plantations, we observed some trees broken by the wind and at plantation C, three out of 40 plants observed had an average windshake observed above bark of 1.3 m length (Figure 7). Finally, some of the trees grown on the south-exposed edge of plantation C had evident sun scalds of the rhytidome (Figure 7).

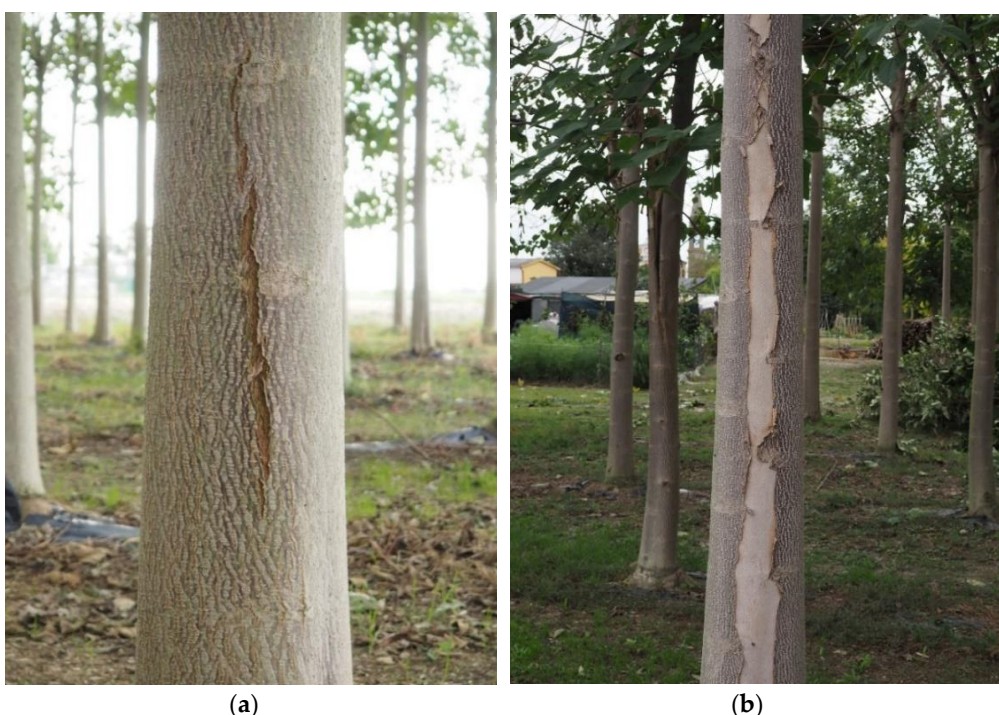

**Figure 7.** Wind shakes (**a**) and sun scald of the rhytidome (**b**) observed at plantation C.

### 3.3. Mass and Volume of Fresh Small Logs

Roundwood density, bark included, was $659 \pm 49$ kg m$^{-3}$ for plantation A, $589 \pm 12$ kg m$^{-3}$ for plantation B, and $531 \pm 94$ kg m$^{-3}$ at plantation C, with an average value of 593 kg m$^{-3}$ (Table 2).

### 3.4. Quality Assessment

According to the EN 1316-2:2012 standard, all 12 selected trees resulted in class Po-A (the best possible quality class for poplar wood) for all measured characteristics except diameter, which resulted in class Po-B and Po-C.

## 4. Discussion

Crown insertion height was around 0.5 m higher at plantation B compared to A and C (Table 2). This parameter was influenced by human activities such as pruning. An early pruning during the first year of growth was carried out in all the plantations, but at plantation B a second pruning was also carried out at the end of the first growing season. Paulownia trees have not shown any problems with producing clear wood up to the height at which they are pruned early (5 m). This pruning height is comparable with the minimal height suggested by [29] and could be increased without any problems to 6–7 m.

Height measured in stand C, where trees were 6 years old, was higher than what was measured in *Paulownia elongata* agroforestry systems in China, where 7-year-old trees are about 8 to 12 m high [30,31]. Based on the number of annual rings observed at the base and at the top of the stem (Table 2), we can conclude that the plants reached 5 m of height during the second vegetative season. This growth is almost double compared to previous experiments with Paulownia Shan Tong (*P. tomentosa x P. fortunei*) grown in Poland between 2016 and 2017 [32], but it is in line with the results of a field experiment in Albania [22] where *Paulownia tomentosa* trees reached a height of $4.25 \pm 0.2$ m at the end of the first growing year after being coppiced.

The diameter measured at breast height, the large, and the middle diameters were very similar at the three plantations despite the different ages of the trees (Table 2). From these data it is clear in this experiment that plant age has a limited influence on plant diameter, suggesting that the growth rate decreases over time. This was confirmed by the significant

decrease in the average annual growth rate with the age of the plantation (Figure 2) and by the significant decrease of ring thickness after the second (plantations A and C) or third year (plantation B) (Figure 4). Our hypothesis is that, with the current plant spacing (3.9 m × 3.9 m at plantation A, 3.8 m × 3.8 m at B, and 4 m × 4 m at C), around the third year of growth the trees' crown expansion results in a competition for light and the competition becomes stronger in the following years, resulting in a sharp decrease in the annual growth rate. This hypothesis has been confirmed by the diameter measured at breast height on interior plants, which was significantly smaller than the diameter measured on edge plants, better exposed to light (Table 2). This is also shown in Figure 6, where interior plants showed thicker rings after the second year of growth compared to edge plants. An example is given in [22] from Western Australia, Queensland, where the plantation scheme ranges from 6 m × 5 m to 6 m × 7 m, and rarely 7 m × 7 m. The density is even lower in the case of agroforestry in China, where *Paulownia elongata* is grown in combination with wheat, with tree density ranging from 45 to 120/hectare [30,31] resulting in plantation schemes ranging from 15 m × 15 m to 9 m × 9 m. Recent indications of how to produce Paulownia in Maryland, USA, reports a plant density of 300–680 trees/acre (740–1680 trees/ha) and a plant spacing of 10 feet by 10 feet (around 3 m × 3 m) [6]. The low spacing proposed by some authors is explained by [6,9], who highlight a particular interest in Paulownia wood with narrow rings for the production of musical instruments to be sold in the USA and Japanese markets. In [9], boards with at least 8 rings per 2.5 cm are described as "high quality wood" and the ones with fewer than 4 rings per 2.5 cm as "low grade wood".

On the other hand, plants placed at the edge of plantation C, mainly those exposed to the south, showed sun scalds of the rhytidome (Figure 7), but no other major defects. In fact, taper was not significantly different in interior plants compared to plants at the edge of the plantation (Table 2).

The plants of the three plantations studied were early and carefully pruned so they did not show knots or significant traces of knots on the shaft. Only in stand B, where the pruning was performed up to about 5 m in the first year and up to about 6 m in the second, was the presence of covered nodes observed.

In all three plantations, some plants broken by the wind were observed. The authors formed the hypothesis that this phenomenon was due to the relatively small diameter compared to the plants' height, due to the high competition for light in the plantations. This characteristic, combined with the low density of Paulownia wood, implies low absolute mechanical performance, resulting in breakages. As suggested by [33], Paulownia trees should be grown in areas with low wind speed (no more than 28 km/h).

The roundwood density observed in this work is a key parameter used to convert wood mass into cubic meters, and vice versa. Usually, 1 m$^3$ of fresh roundwood is considered to weigh approximately 1000 kg, but fresh Paulownia roundwood had a density, on average, of 593 kg m$^{-3}$.

*Quality Assessment*

Quality assessment according to the EN 1316-2:2012 standard has shown a very high quality for Paulownia wood, suggesting that it can be used in the poplar value chain. The plants have shown to be of the best quality class (Po-A) for all the parameters except diameter, which placed the logs in lower quality classes (Po-B and Po-C). Among plant characteristics, diameter is one that can be easily improved by a longer life cycle or increased plant spacing. Because of its low density, Paulownia wood could be used together with or as a substitute for poplar wood for most of the applications that involve it today, such as plywood, edge-glued panels, blockboards, batten boards, and, perhaps in the future, structural glued products.

## 5. Conclusions

*Paulownia elongata x fortunei* (BIO 125 clone) grown in three plantations in Northern Italy has shown outstanding growth performance in terms of diameter and height. Three

years after coppicing, the diametric growth was higher than in previously published literature. At the end of the first year, plants were so tall that farmers could prune them up to 5 m. These features well-describe a fast-growing species able to produce defect-free logs. Growth performance after the third year was significantly reduced for plants located in the interior portion of the stands, highlighting that the spacing applied at the plantations (4 m × 4 m) leads to competition between plants. The need for a larger planting spacing is confirmed by the significantly higher diametric growth of marginal plants after the third year, compared to the interior ones. Paulownia wood has shown an excellent quality ranking in terms of defects and, if classified according to the existing rules for poplar roundwood (another low-density fast-growing species cultivated in northern Italy), it would be in the best class (Po-A) for all features except diameter. This quality ranking confirms the high quality of Paulownia wood harvested in plantations in northern Italy and the need for greater planting spacing to foster diametric growth after the third year. The role of Emilia-Romagna pedoclimatic conditions and silvicultural techniques will be further investigated in future publications.

**Author Contributions:** Conceptualization: G.G. and M.B.; methodology: G.G. and M.B.; sampling and field campaign: I.C., G.G. and M.B.; data analysis: I.C.; writing—original draft preparation: I.C.; writing—review and editing: I.C., G.G. and M.B.; funding acquisition: G.G. and M.B.; supervision: G.G. All authors have read and agreed to the published version of the manuscript.

**Funding:** This research was funded by the Paulownia project (Paulownia: sostenibilità ambientale ed economica per un nuovo sistema forestale, domanda n° 5111574, PSR 2014-2020 Regione Emilia Romagna—Tipo di Operazione 16.1.01 "Gruppi Operativi del PEI per la Produttività e la Sostenibilità dell'agricoltura". Deliberazione della Giunta Regionale N. 2144/2018) funded by the European Agricultural Fund for Rural Development (EAFRD), the Italian Government, and the Emilia Romagna region.

**Data Availability Statement:** Data available on request.

**Acknowledgments:** The authors would like to thank the Paulownia growers' network "PAULOWNIA-Crescere in Rete" for providing access to the stands and making the plants available and C.L.A.F.F. AMBIENTE s.c.a. for leading the project.

**Conflicts of Interest:** The authors declare no conflict of interest.

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
