# Peer review of "Characterization of Paulownia elongata x fortunei (BIO 125 clone) Roundwood from Plantations in Northern Italy"

_forests, doi:10.3390/f13111841_

Round 1

Reviewer 1 Report (Previous Reviewer 2)

The manuscript entitled „Characterization of Paulownia elongata x fortunei (BIO 125 clone) round wood from plantations in Northern Italy” is a case study focusing on assessing the round wood quality of Paulownia cultivated in Northern Italy.

Although interesting from the forest management perspective, the manuscript still lacks scientific discussion of the obtained results. Since this is a scientific article, I strongly suggest adding some scientific explanations.

I had a chance to review the previous version of the paper, and I can see that the Authors did not correct or supplement the manuscript as recommended. Therefore, most of my previous comments are still valid. They can be found in the attached pdf file.

Author Response

see the attached file.

Reviewer 2 Report (New Reviewer)

I think it is an interesting work suitable for the journal. However I have some major concerns which should in my opinion be addressed before publication. Firstly, the manuscript should be presented in a more scientific way, mostly referring to the research hypotheses/questions behind the experimental design. Authors should clearly explain why they carried out a given analysis/comparison linking this with the goal of the study. I have also several important concerns regarding statistical analysis. Also reference list should be in my opinion updated. However I appreciated the efforts of the Authors and I think the work is interesting, therefore, notwithstanding the serious lacks which I found I would like to suggest a major revision.

Line 39: it is not clear what do Authors refer to with “high effort in cultural treatments”, please precise

Introduction section is well written, properly structured and clear but, being Forests a scientific Q1 journal, I think it is worth to present the study aims as research hypotheses or research questions. The objective is clear presented (lines 87-92) but in my opinion restructuring it as: “the aim of the study was to test the hypotheses… or to answer the research questions….” would be more suitable to a scientific journal.

Sub-section 2.1: I suggest to collect the main characteristics of the three study sites in a table, also a map with the localization of the study sites is welcomed. 

Sub-section 2.2: for each of the investigated variables please give the measure unit. Moreover the concept reported in lines 150-152 is not clear. Authors measured the dendrometric features of 20 trees per plantations and then another 20 trees only from the edge of plantation C to represent  an experimental control. But my questions are: why control only in plantation C? And why to measure trees from the edge? From the forestry point of view it is well known that trees growing at the edge of plantations, as a consequence of different light and also edaphic conditions, have different characteristics, like for instance larger diameters and lower height.

Sub-section 2.3: if I understand correctly the only inferential statistic carried out by Authors is between dendrometric characteristics of trees internal to plantation C and trees at its edge. Firstly I still have doubts about the interest for this comparison, please explain why Authors think it is important to include this kind of evaluation in this work. Secondly, if there is only a couple comparison, why ANOVA and not T-test? Finally, Authors referred to Levene test for data homoschedasticity checking but what about data normality?

Lines 234-235: as written before, it is rather predictable that plants growing at the edge of plantations have larger diameters

Figure 1: now it appears a statistical comparison between the three plantations while before it was stated that the aim was not to compare them. Please correct M&M section explaining in a clear way what you are comparing with what and mostly with what aim you are doing this. Moreover I think that the proper measure unit for a growth rate is cm/year not cm. ANOVA analysis is suitable in such case.

Figure 2: Please correct the measure unit as suggested for fig.1. Moreover I suggest to use T-test, not ANOVA as there is a comparison between only two groups.

Figure 3: another statistical analysis and result reported without being described in M&M. Now Authors reported a comparison of ring thickness within the three different plantations. So if I understand correctly the Authors measured the tree rings in some individuals for each plantation and then compared the thickness just between the same plantation (three separated ANOVA analysis one for each plantation). However, if it is like this, the statistic used seems not to be correct. The fact that the Authors are measuring tree rings in the same individuals, i.e. taking tree n. X and measuring its 1st, 2nd, 3rd ring etc…, doing this for N trees within the same plantation and then comparing the obtained data, alters the degrees of freedom and therefore one-way ANOVA is not suitable because data are not independent (ring thickness of 2nd year in tree X is not independent from thickness in 1st year). Therefore the proper statistical analysis is a generalized linear mixed model identifying “tree” as a random factor. This is a serious lack in the framework of data analysis, please redo the analysis or explain clearly why you carried out a simple one-way ANOVA and why you think it is the correct test.

Discussion and conclusion are fine but need to be revised after eventually repeating the statistical analysis.

References: many cited documents are not in peer reviewed journals and/or rather dated. If possible please update the reference list with more recent citations from international peer reviewed journals.

Round 2

Reviewer 1 Report (Previous Reviewer 2)

The manuscript has been corrected.

Reviewer 2 Report (New Reviewer)

I am very satisfied by the way in which authors addressed all my comments and I now propose the acceptance of the manuscript.

This manuscript is a resubmission of an earlier submission. The following is a list of the peer review reports and author responses from that submission.

Round 1

Reviewer 1 Report

Line 13 - Could you please specify the published literature you are referring to? There is no literature mentioned in the Discussion about this topics.

14 - The rate of the further growth is not known due to the fact that 2 of the planting is only 3 or 4 years old.

18-19, 95, ..... - Standards should be listed also in the Reference.

64-65 - Missing in the References and duplicate in this line.

182 - Missing legends.

247 (Density of fresh......) - Missing unit of measures.

268 - The value of “c” on Figure 1. is higher then 4.2.

297-298; 310 - There is no scale on the pictures. Due to that evaluation becomes subjective.

303 - Is Figure 4. necessary? Figure 3. is already showing it.

402 - Missing from the References.

441-443 - It is greatly influenced by other features not mentioned in this publication.

461 - More features of the wood should be known and described to be able to declare that.

509-510 - Missing from the text.

Reviewer 2 Report

The manuscript entitled „Characterization of Paulownia elongata x fortunei (BIO 125 clone) round wood from plantations in Northern Italy” is a case study focusing on the assessment of the round wood quality of Paulownia cultivated in Northern Italy. Although interesting from the forest management perspective, the manuscript lacks scientific discussion of the obtained results. Since this is a scientific article, I would strongly suggest adding some scientific explanations. More comments on the text are provided in the attached .pdf file.
